# Gas Chromatography–Mass Spectrometry-Based Cerebrospinal Fluid Metabolomics to Reveal the Protection of Coptisine against Transient Focal Cerebral Ischemia–Reperfusion Injury via Anti-Inflammation and Antioxidant

**DOI:** 10.3390/molecules28176350

**Published:** 2023-08-30

**Authors:** Junjie Zhang, Ao Qi, Lulu Liu, Chun Cai, Hui Xu

**Affiliations:** 1Key Laboratory of Chinese Medicinal Resource from Lingnan, Research Center of Chinese Herbal Resource Science and Engineering, School of Traditional Chinese Meteria Medica, Guangzhou University of Chinese Medicine, Ministry of Education, Guangzhou 510006, China; 2Southern Marine Science and Engineering Guangdong Laboratory, Zhanjiang 524023, China; 422202a242m.cdb@sina.cn (A.Q.); llulu0413@163.com (L.L.); 3School of Pharmacy, Guangdong Medical University, Zhanjiang 524023, China

**Keywords:** cerebral ischemia–reperfusion, Coptisine, arachidonic acid metabolomics, inflammation, oxidative stress

## Abstract

Coptisine (Cop) exerts a neuroprotective effect on central nervous system disease, particularly ischemic stroke. However, its protective mechanism is still unclear. This study aimed to investigate the protective effect of Cop on cerebral ischemia–reperfusion (IR) rats with a middle cerebral artery occlusion model by integrating a gas chromatography–mass spectrometry (GC–MS)-based metabolomics approach with biochemical assessment. Our results showed that Cop could improve neurobehavioral function and decrease the ischemia size in IR rats. In addition, Cop was found to decrease inflammatory mediators (e.g., prostaglandin D2 (PGD2) and tumor necrosis factor-α (TNF-α) and attenuate oxidative stress response (e.g., increase the superoxide dismutase (SOD) expression and decrease 8-iso-PGF2α level). Furthermore, the GC-MS-based cerebrospinal fluid (CSF) metabolomics analysis indicated that Cop influenced the level of glycine, 2,3,4-trihydroxybutyric acid, oleic acid, glycerol, and ribose during IR injury. Cop exhibited a good neuroprotective effect against cerebral IR injury and metabolic alterations, which might be mediated through its antioxidant and anti-inflammatory properties.

## 1. Introduction

Ischemic stroke is the second leading cause of death in people over 60 years [1]. A stroke-induced energy crisis initiates a range of events, including excitotoxicity, mitochondrial dysfunction, increased calcium levels within cells, and the activation of genes that cause apoptosis. The current primary treatment for ischemic stroke is thrombolysis, which restores blood flow by dissolving the clots and preserving the surrounding brain tissue [2]. However, thrombolytic therapy is unable to reverse ischemic neuronal death and is associated with a number of side effects, including impaired consciousness and speech and a significant reduction in exercise capacity and treatment duration. Therefore, the need for more effective treatment and its corresponding mechanism has been highlighted.

Metabolomics, a rapidly growing technology characterized by the quantitative measurement of dynamical metabolic responses to pathophysiological stimuli or genetic alterations, offers new insight into the mechanisms of disease as well as an assessment of the efficacy of pharmacological treatments. Recently, a liquid chromatography-tandem mass spectrometry-based metabolomic approach was applied to investigate the protective mechanism of Bai-Mi-decoction on ischemic stroke rats and found that 30 differential metabolites were restored by Bai-Mi-decoction intervention, e.g., phenylalanine, valine, and LysoPI(18:0/0:0) [3]. Similarly, when the mechanism of edaravone against cerebral ischemia was investigated by an untargeted metabolomics approach, distinctly different metabolites were screened for their involvement in the biosynthesis of valine, leucine, and isoleucine, as well as in the metabolism of phenylalanine, taurine, and hypotaurine [4]. Lin et al. applied a metabolomic technique to find that Gastrodia elata Blume prevents cerebral IR injury by affecting the following metabolism pathways, including phenylalanine, pyrimidine, and sphingolipid metabolism [5]. Hence, metabolomics techniques are increasingly being used to study the protective mechanisms of pharmacological interventions for ischemic stroke.

The Cop is an effective component of the traditional Chinese medicine Coptidis Rhizoma, which exerts various pharmacological effects such as neuroprotection [6], anti-inflammatory [7,8], antioxidative stress [9,10], and anti-cancer [11,12]. The effects of Cop on central nervous system disorders have recently attracted much interest. The neuro-defensive effects of Cop against excitotoxicity compound glutamate-induced PC12 cell damage and reduction in brain IR injury in rats were reported by Huang et al. [13]. In SH-SY5 cells exposed to tert-butyl hydroperoxide, the neuroprotective effects of Cop were directly correlated with improved mitochondrial membrane potential and reduced apoptosis [14]. In addition, Cop operated through voltage and receptor calcium channels impeded extracellular calcium flooding, which alters a wide range of cellular functions and inevitably reduces neuronal survival [15]. Yu et al. demonstrated that Cop reduced neuronal loss and inhibited the activation of microglia and astrocytes, both of which are involved in the secretion of pro-inflammatory mediators [16]. These findings suggest that Cop may be a promising therapeutic agent for the treatment of central neurological diseases, e.g., stroke. However, more research into the molecular mechanisms of Cop against cerebral IR damage is required. 

In this study, we used middle cerebral artery occlusion to establish IR injury and utilized GC–MS to detect untargeted metabolomics in the CSF between the control, cerebral IR injury, and Cop-treated groups. CSF is a biological fluid that circulates in the brain, It is in direct contact with the extracellular space of the brain, and its changes reflect different functional states of the brain. The aim of our study was to reveal the molecular regulation mechanism of Cop in alleviating cerebral IR injury in rats by biochemical and untargeted metabolomic analysis.

## 2. Results

### 2.1. Protective Effect of Cop on Cerebral IR

The experimental design of our study is depicted in Figure 1A. To confirm whether the cerebral IR rat model was successfully established, 2,3,5-triphenyltetrazolium chloride staining and neurological deficit scores were performed with or without middle cerebral artery occlusion. Figure 1B clearly shows that there were no cerebral infarct lesions in the control group. However, in the cerebral IR group, the infarct size significantly appeared (19.40 ± 2.47%, *p* < 0.001). Meanwhile, the neurological deficit scores in the model group (Figure 1D) were increased in comparison with the control group (*p* < 0.01). Our result suggested that the cerebral ischemic injury rat model was successfully established. 

Subsequently, to investigate the protective effect of Cop against cerebral IR injury, we calculated cerebral infarct size and neurological deficit scores with or without being Cop-treated. A remarkable reduction in infract size (pale-colored region) was illustrated in Cop-treated (2 mg/kg) relative to the model (in Figure 1B), and the cerebral infarct size was significantly diminished after Cop treatment (7.65 ± 2.50%, *p* < 0.01 vs. model group). Neurological functional deficits were evaluated 72 h after surgery in rats, which resulted in the prevention of these deficits by Cop treatment (Figure 1D). These findings indicated that Cop protects against IR-induced brain injury in experimental animals. 

### 2.2. Antioxidant and Anti-Inflammatory Effects of Cop in Response to Cerebral IR 

Enhancement in inflammation leads to cerebral tissue injury. To evaluate the anti-inflammatory effect of Cop during cerebral IR injury, the brain levels of PGD2 and TNF-α were determined. In comparison with the control group, high levels of PGD2 and TNF-α displayed in the cerebral IR group, but they were reversed when treated after Cop (Figure 2). In addition, upregulation in oxidative stress causes oxidative damage to proteins, lipids, and nucleic acids and subsequently results in cell and tissue dysfunction and cell death. Compared with the control, the brain 8-iso-PGF2α in the cerebral IR group significantly increased to 0.113 ± 0.017 μg/mL, while a decrease in it was observed after Cop treatment (0.047 ± 0.021 μg/mL). In contrast, reduced SOD illustrated in the cerebral IR group compared with the control (107.80 ± 10.05 U/mL vs. 68.80 ± 3.48 U/mL, *p* < 0.01), but it was reversed when IR followed by Cop exposure (81.24 ± 2.19 U/mL, *p*-value < 0.05). 

### 2.3. GC–MS Analysis

In this study, we applied an untargeted GC–MS approach to investigate the CSF metabolic profile among control, cerebral IR, and Cop-treated groups. From our result, more than 50 compounds in the CSF sample were separated and identified, as shown in Appendix A. Quality control (*n* = 6) was tested to verify the reliability of the method before the actual samples were measured. Six compounds (including amino acids, sugars, and fatty acids) in the quality control samples were selected as targets of the method, which had different charge-to-mass ratios, chromatographic retention time, and satisfying conditions for better peak shape and higher peak intensities. As can be seen in Appendix A, the relative standard deviation values of the retention time and peak intensity were in the range of 0.013–0.037% and 2.584–5.873%, respectively. Our data indicated that the method has good reproducibility and stability. 

### 2.4. Screen in Metabolites from GC–MS Data as Potential Biomarkers between Control, Model, and Cop-Treated Groups

To initialize the systemic alteration of metabolites, a multivariate pattern discriminative analysis was conducted. A partial least square discriminant analysis (PLS-DA) score plot of GC–MS data in CSF illustrated that control and cerebral IR injury could be well differentiated (Figure 3A), implying that the metabolic profile was significantly altered after cerebral IR. Meanwhile, Cop-treated and untreated rats were always separated clearly (Figure 3B). Our results suggest that Cop could be useful in the treatment of cerebral IR injury.

Subsequently, potential targets between cerebral IR and Cop-treated were screened by *p*-value < 0.05 in the *t*-test and |FC| > 1.5 (Figure 4A,B). Metabolite levels in response to cerebral IR and Cop-treated were visualized in a heatmap that metabolites with warm color represented up-regulation and with cool color reflected down-regulation. There were 47 differential metabolites discovered between IR and control groups in CSF, including 16 increased (ribose, ethanedioic acid and butanoic acid, etc.) and 31 decreased (glycine, oleic acid, gluconic acid and cadaverine, etc.), otherwise there were 6 differential metabolites in CSF (4 increased (oleic acid, glycerol, glycine, and 2,3,4-trihydroxybutyric acid) and 2 decreased (ribose and stearic acid)) after Cop-treated (Appendix A). The volcano plot and heatmap also showed the expression level of differential metabolites among three groups in CSF (Figure 4). As can be seen from the above results, significant differences in metabolites were shown among the three groups. Therefore, the role of Cop in the treatment of cerebral IR injury is closely related to the alteration in expressions of certain metabolites. 

To assess the key metabolites induced by cerebral IR and Cop treatment, we employed a Venn map to identify the shared metabolites in the CSF of rats subjected to cerebral IR and Cop treatment (Figure 5A). From our result, five common metabolites as potential markers were displayed between the control, IR, and Cop-treated, including glycerol, oleic acid, glycine, ribose, and 2,3,4-trihydroxybutyric acid (Figure 5 and Appendix A). 

### 2.5. Analysis of Metabolic Pathway Enrichment

To further explore the mechanisms responsible for cerebral IR and Cop-treated, enrichment and metabolic pathway analyses of markers were conducted using the MetaboAnalyst 5.0 platform. The enrichment of metabolic pathways was screened according to *p* value < 0.05. As can be seen in Figure 5B, CSF biomarkers of model and control enriched in the (1) Phenylalanine, tyrosine, and tryptophan biosynthesis; (2) Galactose metabolism; (3) Pentose phosphate pathway; (4) Glycolysis/Gluconeogenesis; (5) Glutathione metabolism; and (6) Glyoxylate and dicarboxylate metabolism. Additionally, the Glycerolipid metabolism was observed in the enrichment analysis between cerebral IR and Cop-treated.

## 3. Discussion

To our knowledge, this is the first report that investigates the mechanism of Cop against cerebral IR by untargeted CSF metabolomes. In this study, we observed that Cop attenuated cerebral IR-induced cerebral ischemia size by 2,3,5-triphenyltetrazolium chloride staining and decreased neurological deficit function score, suggesting that Cop exerted a neuroprotective role in cerebral IR. To investigate the protective mechanism of Cop against cerebral IR, biochemical indicators and an untargeted metabolomic experiment were carried out. In the process of cerebral IR, excessive release of pro-inflammatory factors PGD2 and TNF-α increases blood–brain barrier permeability, promoting the infiltration and aggregation of leukocytes, thereby causing neuroinflammation [17,18]. Our data revealed that Cop decreased the level of PGD2 and TNF-α in the brain. Moreover, imbalances in oxidative stress-induced cerebral IR injury were reported by previous studies. Changes in SOD and 8-iso-PGF2α present the oxidative stress state in vivo [19,20,21]. Our result showed that upregulation on 8-iso-PGF2α and downregulation on SOD expression displayed in cerebral IR compared with control, whereas their levels were reversed through Cop treatment. These results indicated that antioxidant and anti-inflammation effects are the mechanism of Cop against cerebral IR.

A large body of literature reports about the metabolic changes after cerebral IR injury and attempts to find the corresponding potential therapeutic targets. However, there are relatively few studies on the response of the CSF metabolome to Cop during cerebral IR therapy. In our results, CSF five compounds (glycine, glycerol, oleic acid, 2,3,4-trihydroxybutyric acid, and ribose) were discovered by untargeted metabolomic analysis. Interestingly, the role of these analytes in vivo has been reported to be closely associated with anti-inflammation and antioxidants. 

Glycine, a nerve-inhibitory amino acid, exerts neuroprotective properties. Goulart and their colleagues also reported that diminished glycine was observed in ischemic stroke compared with control [22], which is similar to our studies. Supplement on glycine reduces the secretion of pro-inflammatory cytokine in PC12 cells with induction of oxygen–glucose deprivation [23]. Liu et al. demonstrated that a glycine-treated cerebral IR injury rat was able to suppress ischemia-mediated inflammation by improving M2 microglia polarization and inhibiting M1 microglia polarization by the NF-κB p65/Hif-1ɑ signaling pathway [24]. Our result showed that a high level of glycine was observed after Cop treatment. 

Oleic acid, a principal component of phospholipids, is abundant in myelin sheaths. Recent studies have shown that oleic acid plays a crucial role in the normal development and function of the brain, acting as a neurotrophic factor that contributes to the growth of axons and dendrites, enhancement of neuronal migration and aggregation, and promotion of synapse formation [25]. Vascellari et al. reported that a decrease in oleic acid levels was observed in Parkinson’s disease compared with healthy individuals [26]. In vitro, oleic acid has been shown to reduce the inflammatory response of microglia, which leads to neuronal death after cerebral ischemia [27]. Song et al. reported that oleic acid significantly reduced the infarct volume after cerebral ischemia, prevented CA1 neuronal death, and markedly attenuated the immunoreactivity of cyclooxygenase-2, inducible nitric oxide synthase and TNF-α [28]. Our data indicated that down-regulation in oleic acid level was illustrated in cerebral IR injury compared with control, while upregulation in it was observed when treated after Cop. 

As an aldehyde containing pentose sugar, ribose participates in numerous biochemical processes and has an active role in the glycation of protein, producing advanced glycation end products that have severe cytotoxicity, which can lead to cell dysfunction and death. Zhang found that ribose decreased cell viability and increased lactate dehydrogenase release in mesangial cells, leading to glutathione depletion, a naturally occurring antioxidant present in cells, and clear intracellular reactive oxygen species [29]. Yu et al. reported that decreased ribose markedly reduced advance glycation end-product accumulation, Tau hyperphosphorylation, and neuronal death [30]. Moreover, ribose-activated NLRP3 inflammasome and an induced NF-κB pathway via the advance glycation end products pathway was evidenced by Hong et al. [31,32]. NLRP3 inflammasome facilitates pro-inflammatory cytokine secretion by mediating caspase-1 activation. Wu et al. reported that Cop not only hindered the assembly of NLRP3 inflammasome but also suppressed the initiation of inflammasome by inactivating the NF-κB pathway to reduce the expression of NLRP3 [33]. Xiong et al. reported that Cop decreased the level of TNF-α, IL-1β, and IL-18 and significantly downregulated the protein expression levels of NLRP3, apoptosis-associated speck-like protein containing a CARD and caspase-1 [34]. Our results showed that ribose levels were increased in the cerebral IR group and that Cop reversed it. We hypothesized that Cop might reduce ribose levels to inhibit NF-κB inflammatory signaling caused by the activation of the NLRP3 inflammasome, and a specific mechanism needs to be further studied. 

Glycerol is a precursor for membrane phospholipid synthesis and the main lipid constituent of the myelin sheath. Stroke is associated with the disruption of the blood–brain barrier, as evidenced by enhancing matrix metalloproteinase-9 activity and reducing zonula occludens-1 protein expression. Chang et al. reported that glycerol reduced matrix metalloproteinase-9 activity and increased zonula occludens-1 protein expression in the stroke model [35]. Meanwhile, glycerol could also alleviate oxidative stress and neuroinflammation in stroke by the NF-κB pathway. Previous studies have demonstrated that the increase in glycerol and the decrease in formate levels were detected in the ischemic brain treated with a drug and that drug-mediated increase in glycerol was associated with myelin formation in oligodendrocytes, which serve to maintain and protect normal neuronal function [36]. Cop increased glycerol levels in CSF, suggesting that it might be associated with anti-inflammation. 

The 2,3,4-trihydroxybutyric acid (namely erythronic acid), which is commonly found in the aqueous humor of the eye and in cerebrospinal fluid, is derived from glycated proteins. Our result revealed that a lower level of it was displayed in the cerebral IR group. Lower blood levels of 2,3,4-trihydroxybutyric acid presented in patients with acute ischemic stroke compared with healthy patients [37], which is similar to our results. The role of 2,3,4-trihydroxybutyric acid in brain injury is still unclear, but it might be negatively associated with inflammation and oxidative stress. Huang et al. reported that the level of 2,3,4-trihydroxybutyric acid in hyperlipidemia acute pancreatitis decreased observably compared with the health control [38]. Moreover, 2,3,4-trihydroxybutyric acid is an oxidative product of erythritol, which has excellent hydroxyl radical scavenger properties. Jin et al. reported that erythritol reversed lipid accumulation, promoted the production of heme oxygenase 1 and NQO1 antioxidant proteins, and inhibited the expression of endoplasmic reticulum stress proteins GPR78, p-PERK, and CHOP by the nuclear factor E2-related factor pathway [39]. Both erythritol and 2,3,4-trihydroxybutyric acid have been identified as products when N-acetylglucosamine is oxidized NaOCl, which relates to reactive oxygen species degradation of connective tissue [40]. In our result, Cop treatment increased the level of 2,3,4-trihydroxybutyric acid. We hypothesize that the regulation mechanism of Cop might be associated with antioxidants. Cop significantly promotes the activation of nuclear factor-erythroid-2-related factor 2 and heme oxygenase-1, thereby inhibiting H2O2-induced cytotoxicity and DNA damage [9]. Zhai and their colleagues also illustrated that the antioxidative effect of Cop was linked with upregulation in the expression of nuclear factor-erythroid-2-related factor 2 and NADPH quinone oxidoreductase 1 [41].

## 4. Materials and Methods

### 4.1. Chemical Reagents 

Methanol was chromatographic grade and obtained from Merck (Darmstadt, Germany). Cop chloride (purity > 98%) was purchased by Yirui Biotechnology Co., Ltd. (Chengdu, China), and Methoxyamine hydrochloride and N, O-bis (trimethylsilyl) trifuoroacetamide (1% trimethylchlorosilane) were supplied by Macklin (Shanghai, China). Heptadecanoic acid (purity > 98%) as an internal standard was obtained from Sigma-Aldrich (St. Louis, MO, USA) and dissolved with ethyl acetate at 100 μg/mL. A Milli-Q purification system (Billerica, MA, USA) was used to get ultrapure water.

### 4.2. Animal Care

The experimental protocol was in accordance with internationally accepted standards and was approved by the Ethics Committee for Guangdong Medical Laboratory Animal Center (Foshan, China). Eighteen male Sprague–Dawley rats (300–320 g) were furnished by Guangdong Medical Laboratory Animal Center and maintained under controlled circumstances with a 12 h light–dark at 21 ± 2 °C and 60% ± 5% (relative humidity) for at least five days before the experiment. Standard rodent chow and tap water were freely accessible for rats. 

### 4.3. Transient Focal Cerebral IR and Animal Treatment

The intraluminal suture method was used to close the middle cerebral artery. First, rats were sedated with chloral hydrate (350 mg/kg, i.p.). The origin of the middle cerebral artery was then blocked with a 3-0 monofilament after 2 cm skin dissection at the ventral muscle of the rat’s neck. After 60 min of ischemia, the monofilament was removed, and reperfusion was completed, followed by wound closure. During the operation and reperfusion, a heated blanket kept the rectal temperature at 37 ± 0.5 °C.

After surgery, rats were randomized into two groups: IR (*n* = 6) and IR after Treatment (*n* = 6). After 60 min of reperfusion, rats were given an intraperitoneal injection of Cop (2 mg/kg/qd), while the control (Sham, *n* = 6) and IR groups received saline instead.

### 4.4. Neurological Function Measurement 

The neurologic function of conscious rats was blindly evaluated 72 h after IR injury. Neurological deficits scores in rats were assessed by an observed blinded to the group using a previously reported five-point scale [42]. The higher the score is, the more severe the neurological deficit is.

### 4.5. Sample Collection 

After 96 h of cerebral IR injury, rats were given chloral hydrate (350 mg/kg,i.p.) for anesthesia, then placed on a homemade device that bent their heads downward 45° and a small midline incision between the ears was made to separate the muscles to expose the atlanto-occipital membrane located between the occipital bone and the upper cervical vertebrae. A butterfly needle (25 gauge × 19 mm) was used to aspirate clear CSF (nearly 50 μL) into the syringe until blood appeared, and the clear CSF was transferred to an Eppendorf tube and stored at −80 °C. Brain tissue (*n* = 6/per group) was collected and cut into six coronal sections of 2 mm thickness; finally, these sections were stained with 2% 2,3,5-triphenyltetrazolium chloride to assess the size of cerebral IR injury. 

### 4.6. Sample Preparation and GC–MS Analysis

First, 20 μL CSF, 15 μL heptadecanoic acid, and 1mL methanol were mixed and then vortexed for 10 min at 4 °C, and the supernatant was extracted by centrifugation at 15,000 rpm for 5 min and subsequently dried with nitrogen. The residue in the vial was dissolved with 50 μL of 2% methoxyamine hydrochloride and incubated at 37 °C for 90 min, and then 50 μL of N, O-bis (trimethylsilyl)trifluoroacetamide (1% trimethylchlorosilane) was added and reacted for 60 min at 70 °C. To prevent metabolite degradation, samples were stored at −80 °C prior to analysis.

The GC–MS-QP2010 Ultra (Shimadzu, Japan) with electron impact mode was employed to analyze the CSF metabolites, and to isolate compounds, a DB-5 capillary column (30 m × 0.25 mm × 0.25 μm, Agilent, Santa Clara, CA, USA) was used. Each sample was subjected to an injection volume of 1 μL, and the injection mode was set to splitless. The temperature program was listed as follows: kept at 85 °C for 2 min, increased to 150 °C at 5 °C/min, then to 240 °C at 15 °C/min, followed by 280 °C at 40 °C/min, and finally held at 280 °C for 5 min. The temperature of the inlet, transfer line, and ion source were set at 230 °C, 250 °C, and 200 °C, respectively. The carrier gas used was helium (99.999%), with a set pressure of 100 kPa and constant-flow control mode. The full scan mode was used to conduct the analyses and set the monitoring mass range of MS to 50–650.

### 4.7. Qualitative Identification of Metabolites and Raw Data Preprocessing

Compounds were identified using the NIST 11.0 library, and qualitative analysis of compounds was done using a minimum of 80% similarity as a screening criterion.

The peak areas of compounds were standardized and corrected by dividing the peak area of heptadecanoic acid, logarithmic transformation, and auto-scaling was used to complete a normal distribution. The corrected peak areas of all identified metabolites were then used for further analysis. 

### 4.8. Measurement of Brain PGD2, TNF-α, 8-iso-PGF2α and SOD 

The levels of PGD2 and 8-iso-PGF2α in the brain were detected by liquid chromatography-tandem mass spectrometry as in previous studies [43]. The level of TNF-α and SOD was calculated by a commercial ELISA kit according to the instruction of the manufacturer (Shanghai Enzyme-linked Biotechnology Co., Ltd., Shanghai, China)

### 4.9. Statistical Analysis

Data were presented as mean ± standard deviation (SD). Metabolomic analysis, including multivariate analysis (principal component analysis and PLS-DA and cluster analysis, was achieved using the online software MetaboAnalyst 5.0 (http://www.metaboanalyst.ca/, accessed on 20 June 2023). To determine statistical significance, we employed the student’s *t*-test (SPSS 19.0 software). When the *p*-value following the Benjamini–Hochberg false discovery rate correction was less than 0.05, the difference was declared significant. 

## 5. Conclusions

In this study, Cop significantly reduced the ischemia size and improved neurological deficit scores after IR injury. Moreover, Cop inhibited the release of pro-inflammatory mediators (e.g., PGD2 and TNF-α) and ameliorated oxidative stress (increasing SOD expression and suppressing 8-iso-PGF2α level). Furthermore, the metabolomic analysis showed different CSF metabolic profiles among the control, IR, and Cop-treated groups. A total of five metabolites in CSF were identified as targets, and the related pathways analysis was further conducted. The combination of biochemical indicator and metabolomics methods revealed that Cop exerted an anti-inflammation and antioxidant effect against cerebral IR injury.

## Figures and Tables

**Figure 1 molecules-28-06350-f001:**
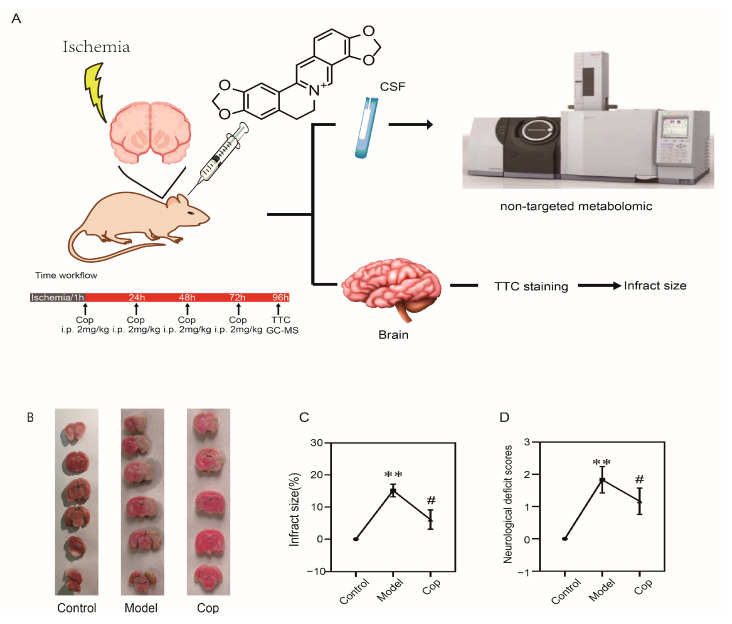
Coptisine (Cop) was effective in protecting the rats with cerebral IR injury. (**A**) Flow chart of the experimental protocol. (**B**) Representative images of the 2,3,5-triphenyltetrazolium chloride stain at 96 h after reperfusion. White color: infract region; red color: non-infract region. (**C**) Statistical results of infract size in rats (*n* = 6/per group). (**D**) Results of neurological deficit score in rats. Data were presented as mean ± SD. (# *p* < 0.05 (model vs. Cop), ** *p* < 0.01 (control vs. model)).

**Figure 2 molecules-28-06350-f002:**
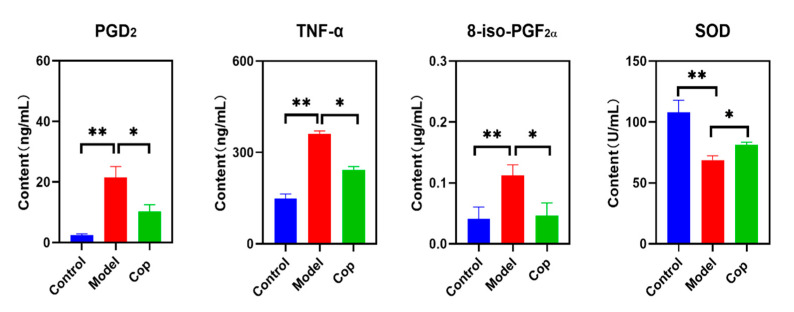
Change in PGD2, TNF-α,8-iso-PGF2α, and SOD among three groups (* *p* value < 0.05, ** *p* value < 0.01).

**Figure 3 molecules-28-06350-f003:**
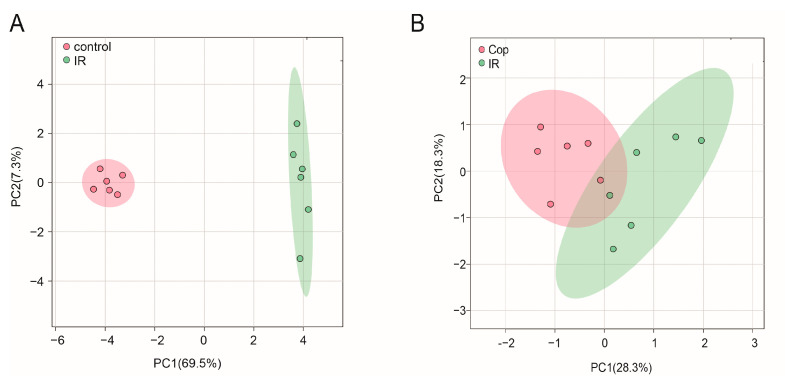
Untargeted metabolomic analysis of CSF after cerebral IR and Cop-treated in rats. Plot of CSF PLS-DA between the control and IR (**A**) group and IR and Cop-treated (**B**) group.

**Figure 4 molecules-28-06350-f004:**
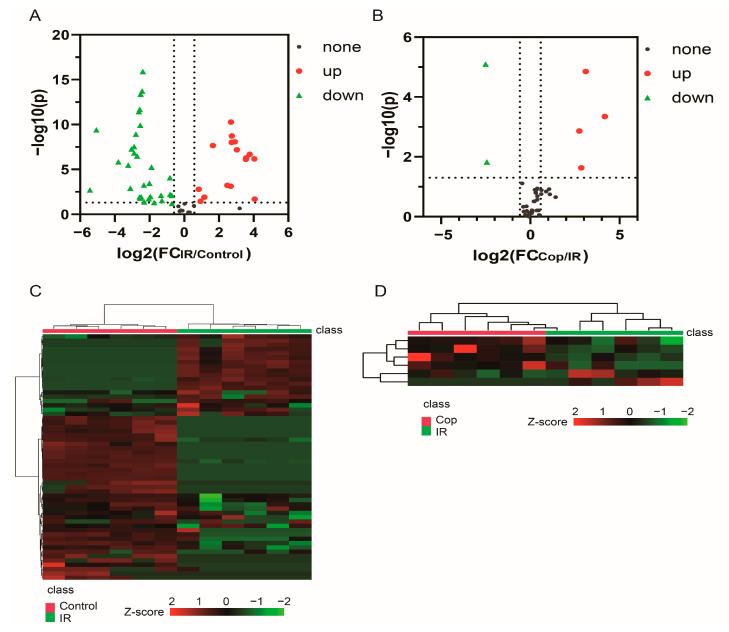
The volcano map and heatmap showed the expression level of differential compounds in CSF. Screen in CSF markers in control vs. IR (**A**) and IR vs. Cop (**B**) according to |fold change| > 1.5 and *p*-value < 0.05. Red color: upregulation; Green color: downregulation. Change in differential metabolites in control vs. IR (**C**) and IR vs. Cop (**D**). Red color indicates upregulation, and green color indicates downregulation.

**Figure 5 molecules-28-06350-f005:**
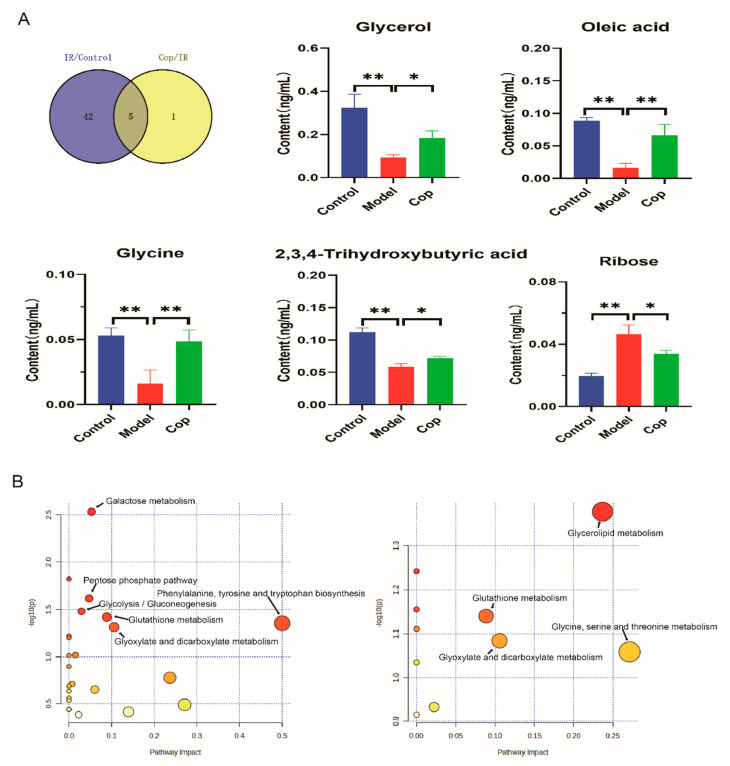
The unique differential compounds of different comparison groups. (**A**) The Venn plot indicated the common different compounds as potential targets among three groups, and the bar chart revealed changes in their level among the three groups (* *p* value < 0.05, ** *p* value < 0.01). (**B**) Enrichment analysis of metabolic pathway of differential compounds between control and IR (**left**), IR and Cop treated (**right**).

## Data Availability

The datasets used and/or analyzed during the current study are available from the corresponding author upon reasonable request.

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
