# Peer review of "Gas Chromatography–Mass Spectrometry-Based Cerebrospinal Fluid Metabolomics to Reveal the Protection of Coptisine against Transient Focal Cerebral Ischemia–Reperfusion Injury via Anti-Inflammation and Antioxidant"

_molecules, 2023, doi:10.3390/molecules28176350_

Round 1

Reviewer 1 Report

I wonder the authors used Coptisine or Coptisine chloride. From Fig1 A, I think it should be Coptisine chloride or other form of salt. Please double-check.

In line 78 and elsewhere in the text, the authors misspelled 2,3,5-triphenytetrazolium chloride, it should be 2,3,5-triphenyltetrazolium chloride.

I am not sure I fully understand the meanings of lines 47 and 283. Please make them clearly for readers.

What are the biomarkers the authors found in this study?

The authors need to carefully check their grammars and spellings.

A lot of nouns should be used plurals instead the authors used singulars.

Author Response

We feel great thanks for your professional review work on our manuscript. As you are concerned, there are several problems that need to be addressed. According to your nice suggestion, we have made extensive corrections to our manuscript, the details corrections are listed below.

  1. I wonder the authors used Coptisine or Coptisine chloride. From Fig1 A, I think it should be Coptisine chloride or other form of salt. Please double-check.

Response: Thanks for your indication. This mistake has been corrected in the section of material and methods. “Cop chloride(purity>98%) was purchased by Yirui Biotechnology Co.Ltd.(Chengdu, China)”

  1. In line 78 and elsewhere in the text, the authors misspelled 2,3,5-triphenytetrazolium chloride, it should be 2,3,5-triphenyltetrazolium chloride.

Response: Thanks for your comments. This mistake has been corrected.

  1. I am not sure I fully understand the meanings of lines 47 and 283. Please make them clearly for readers.

Response: Thank you for your advice. (1) The sentence in line 47 has been corrected as follows: “In recent, a liquid chromatography tandem mass spectrometry based-metabolomic approach was applied to investigate the protective mechanism of Bai-Mi-decoction on is-chemic stroke rats, and found 30 differential metabolites were restored by Bai-Mi-decoction intervention, e.g., phenylalanine, valine and LysoPI(18:0/0:0).” (2) The sentence in line 283 has been corrected as follows: “ Zhai and their colleagues also illustrated that the antioxidative effect of Cop was linked with upregulation in the expression of nuclear factor-erythroid-2 related factor 2 and NADPH quinone oxidoreductase 1. ”. We hope these revisions would make readers understand.

  1. What are the biomarkers the authors found in this study?

Response: Thanks for your advice. The information of CSF biomarkers between control, cerebral IR injury and Cop treated add in the manuscript and supplementary materials (Tab S3, S4). Finally, five CSF metabolites were screen as target of Cop against cerebral IR(Tab S5 and Fig 5A).

Table S1 Retention time and similarity of analytes in SD rats’ CSF by GC-MS

Compound

tR(min)

Similarity (%)

Compound

tR(min)

Similarity (%)

Cyclobutane

2.027

80

Pentanedioic acid

17.315

88

3-Buten-2-ol

2.415

81

Glutamine

17.904

92

alpha-Pinene

3.185

96

Phenylalanine

17.959

95

Ethane

3.225

97

Ribonic acid

18.205

80

2-Thenaldehyde

3.38

81

Dodecanoic acid

18.345

83

Benzene

3.435

86

Lyxose

18.37

86

borate

3.605

93

Ribose

18.4

92

ethane-1,2-diol

3.67

92

Asparagine

18.521

93

Dimethylamine

4.05

93

Arabinose

18.579

89

Pentane

4.655

95

Lysine

18.715

81

Propanoic acid

4.775

93

Sorbitol

18.805

89

Acetic acid

5.315

93

Arabitol

19.035

90

n-Butylamine

5.705

92

Phosphoric acid

19.482

87

Alanine

5.885

92

Galactopyranose

19.94

96

1,4-Butanediol

5.92

84

1,2,3-PA

20.03

86

Butanoic acid

6.415

94

D-Ribo-Hexitol

20.075

85

1,2-Butanediol

6.73

83

Gluconic acid

20.11

83

Ethanedioic acid

6.78

82

Erythrotetrofuranose

20.325

84

Valine

8.628

88

Fructose

20.47

93

m-ethynylaniline

8.7

84

Galactose

20.595

93

Urea

9.415

93

Glucose

20.69

89

Glycerol

10.185

93

Dulcitol

21.01

91

2(3H)-Furanone

10.25

92

Ribitol

21.011

84

Isoleucine

10.676

93

Tyrosine

21.085

87

Threonine

10.695

90

Sucrose

21.18

85

Glycine

10.98

94

Cholesterol

21.266

86

Butanedioic acid

11.295

94

Pantothenic acid

21.45

82

Chlorphentermine

11.373

80

Mannose

21.639

83

2,3-DBA

11.992

91

Palmitelaidic acid

21.711

83

Serine

12.48

92

Hexadecanoic acid

21.825

91

Cadaverine

13.753

84

Inositol

22.026

85

AA

15.322

83

Uric acid

22.079

84

Malic acid

15.762

90

Stearic acid

22.795

84

Threitol

16.13

90

Oleic acid

22.93

89

Proline

16.33

96

2-monopalmitin

23.35

84

Aspartic acid

16.376

90

2-Monostearin

23.745

86

Pentanoic acid

16.545

82

13-Docosenamide

24.195

89

Creatinine

16.87

91

1,2-BA

24.813

86

2,3,4-TBA

17.03

95

Abbreviation:1,2,3-PA:1,2,3-Propanetricarboxylic acid;2,3,4-TBA:2,3,4-Trihydroxybutyric acid;1,2-BA: 1,2-Benzenedicarboxylic acid; 2,3-DBA:2,3-Dihydroxybutanoic acid;

Table S2 Reproducibility of analytes in Quality Control from CSF by GC-MS 

Metabolites

tR(min)

Content(ng/mL)

Mean

SD

RSD(%)

Mean

SD

RSD(%)

Alanine

5.889

0.002

0.037

0.158

0.009

5.873

Serine

12.489

0.005

0.040

0.237

0.007

2.853

Cadaverine

13.754

0.005

0.037

0.058

0.002

4.139

Glucose

20.692

0.003

0.013

10.506

0.475

4.523

Sucrose

21.182

0.003

0.014

0.195

0.007

3.578

Stearic acid

22.794

0.004

0.016

0.063

0.002

2.584

Abbreviation:RSD: relative standard deviation.SD:standard deviation.

Tab S3 Screen in CSF marker between control and cerebral IR

Compound

Log2(FC)

Log10(p)

Compound

Log2(FC)

Log10(p)

Lysine

-5.432

2.703

Arabitol

-1.890

5.254

Propanoic acid

-5.066

9.415

2(3H)-Furanone

-1.737

1.302

13-Docosenamide

-3.794

5.836

Dodecanoic acid

-1.338

2.119

Sucrose

-3.222

5.474

n-Butylamine

-1.288

1.543

Gluconic acid

-3.100

2.913

Acetic acid

-0.821

4.083

2,3,4-TBA

-3.036

7.270

1,2,3-PA

-0.811

2.231

2-monopalmitin

-2.887

7.536

Hexadecanoic acid

-0.712

2.109

D-Fructose

-2.881

6.850

ethane-1,2-diol

0.831

2.794

Dulcitol

-2.782

8.923

Pantothenic acid

0.942

1.452

Threitol

-2.717

6.489

Ethanedioic acid

1.161

1.905

Lyxose

-2.614

11.446

Glucose

1.650

7.664

2-Monostearin

-2.573

1.932

2-Thenaldehyde

2.490

3.221

Ribonic acid

-2.561

11.687

Alanine

2.689

3.122

Glycerol

-2.540

9.927

borate

2.698

10.277

Galactopyranose

-2.540

1.811

Tyrosine

2.744

8.730

Ethane

-2.514

13.378

Ribose

2.914

8.069

Cadaverine

-2.490

1.939

Erythrotetrofuranose

3.032

7.199

Dimethylamine

-2.421

13.737

Butanoic acid

3.193

0.656

3-Buten-2-ol

-2.390

15.900

Creatinine

3.555

6.126

Sorbitol

-2.323

3.251

Serine

3.577

6.291

Oleic acid

-2.309

1.387

Pentane

3.772

6.677

D-Ribo-Hexitol

-2.050

1.723

1,4-Butanediol

4.049

6.168

Glycine

-2.007

3.464

Galactose

4.068

1.691

1,2-Butanediol

-1.957

1.993

Tab S4 The candidate markers of CSF between cerebral IR and Cop treated.

Compound

Log2(FC)

-Log10(p)

Ribose

Stearic acid

Glycerol

2,3,4-TBA

Glycine

Oleic acid

-2.502

-2.432

2.742

0.587

2.852

4.170

4.533

1.830

3.095

1.435

1.834

4.674

Tab S5 Identification in CSF markers between control, cerebral IR and Cop treated groups

Compounds

Control vs Cerebral IR

Cerebral IR vs Cop treated

Log2(FC)

-Log10(p)

Log2(FC)

-Log10(p)

Ribose

Glycerol

2,3,4-TBA

Glycine

Oleic acid

2.914

-2.540

-3.036

-2.007

-2.309

8.069

9.927

7.270

3.464

1.387

-2.502

2.742

0.587

2.852

4.170

4.533

3.095

1.435

1.834

4.674

  1. The authors need to carefully check their grammars and spellings. A lot of nouns should be used plurals instead the authors used singulars

Response: Thanks for your indication. These mistakes have been corrected.

Reviewer 2 Report

The paper entitled “GC-MS based cerebrospinal fluid metabolomics to reveal the protection of Coptisine against transient focal cerebral ischemia-reperfusion injury via ant-inflammation and antioxidant” is a well-designed study that demonstrated the protective effect of Coptisine in the focal ischemia cerebral model in rat. The authors demonstrated that Coptisine reduces the infarct area and partially prevents neurobehavioral dysfunction. The treatment also decreases some inflammatory mediators and attenuates oxidative stress. Additionally, it was demonstrated that 5 metabolites are modulated by the Coptisine treatment. These were identified by the GC-MS metabolomics analysis. Results are clear and consistent. The discussion is excellent, the authors found many related papers that corroborated and support their findings in the metabolomics analysis.

I didn’t have access to supplementary tables.

I don't have important observations, just two comments:

 ·         Definition of CSF is missing in the introduction (line 72), it only appears in the abstract

·         In the description of results obtained in Figure 4, the authors should indicate the panel to facilitate comprehension.

·         Conclusion should be changed before methods

Although in general, the authors use Standard English, there are some words that are not used properly. I suggest reviewing carefully the text. For example:

Line 82: “the cerebral ischemia size was apparently increased in cerebral…” The common form to refer to the damaged area is “infarct area” instead of “ischemia size”. The authors found statistical differences between control and ischemic groups, so, it is not necessary to say “apparently”.

Line 85: “…to investigate whether the protective effect...”. The use of "whether" is incomplete in the sentence.

Lines 142 and 144, have similar errors.

Author Response

We feel great thanks for your professional review work on our manuscript. As you are concerned, there are several problems that need to be addressed. According to your nice suggestion, we have made extensive corrections to our manuscript, the details correction are listed below.

  1. I didn’t have access to supplementary tables.

Response: Thanks for your indication. Due to our negligence, the supplementary tables were forgot to attach. Now we have attached them in our manuscript. The information of CSF biomarkers between control, cerebral IR injury and Cop treated add in the manuscript and supplementary materials (Tab S3 and S4). Finally, five CSF metabolites may be target of Cop against cerebral IR (Tab S5 and Fig 5A).

Table S1 Retention time and similarity of analytes in SD rats’ CSF by GC-MS

Compound

tR(min)

Similarity (%)

Compound

tR(min)

Similarity (%)

Cyclobutane

2.027

80

Pentanedioic acid

17.315

88

3-Buten-2-ol

2.415

81

Glutamine

17.904

92

alpha-Pinene

3.185

96

Phenylalanine

17.959

95

Ethane

3.225

97

Ribonic acid

18.205

80

2-Thenaldehyde

3.38

81

Dodecanoic acid

18.345

83

Benzene

3.435

86

Lyxose

18.37

86

borate

3.605

93

Ribose

18.4

92

ethane-1,2-diol

3.67

92

Asparagine

18.521

93

Dimethylamine

4.05

93

Arabinose

18.579

89

Pentane

4.655

95

Lysine

18.715

81

Propanoic acid

4.775

93

Sorbitol

18.805

89

Acetic acid

5.315

93

Arabitol

19.035

90

n-Butylamine

5.705

92

Phosphoric acid

19.482

87

Alanine

5.885

92

Galactopyranose

19.94

96

1,4-Butanediol

5.92

84

1,2,3-PA

20.03

86

Butanoic acid

6.415

94

D-Ribo-Hexitol

20.075

85

1,2-Butanediol

6.73

83

Gluconic acid

20.11

83

Ethanedioic acid

6.78

82

Erythrotetrofuranose

20.325

84

Valine

8.628

88

Fructose

20.47

93

m-ethynylaniline

8.7

84

Galactose

20.595

93

Urea

9.415

93

Glucose

20.69

89

Glycerol

10.185

93

Dulcitol

21.01

91

2(3H)-Furanone

10.25

92

Ribitol

21.011

84

Isoleucine

10.676

93

Tyrosine

21.085

87

Threonine

10.695

90

Sucrose

21.18

85

Glycine

10.98

94

Cholesterol

21.266

86

Butanedioic acid

11.295

94

Pantothenic acid

21.45

82

Chlorphentermine

11.373

80

Mannose

21.639

83

2,3-DBA

11.992

91

Palmitelaidic acid

21.711

83

Serine

12.48

92

Hexadecanoic acid

21.825

91

Cadaverine

13.753

84

Inositol

22.026

85

AA

15.322

83

Uric acid

22.079

84

Malic acid

15.762

90

Stearic acid

22.795

84

Threitol

16.13

90

Oleic acid

22.93

89

Proline

16.33

96

2-monopalmitin

23.35

84

Aspartic acid

16.376

90

2-Monostearin

23.745

86

Pentanoic acid

16.545

82

13-Docosenamide

24.195

89

Creatinine

16.87

91

1,2-BA

24.813

86

2,3,4-TBA

17.03

95

Abbreviation:1,2,3-PA:1,2,3-Propanetricarboxylic acid;2,3,4-TBA:2,3,4-Trihydroxybutyric acid;1,2-BA: 1,2-Benzenedicarboxylic acid; 2,3-DBA:2,3-Dihydroxybutanoic acid;

Table S2 Reproducibility of analytes in Quality Control from CSF by GC-MS 

Metabolites

tR(min)

Content(ng/mL)

Mean

SD

RSD(%)

Mean

SD

RSD(%)

Alanine

5.889

0.002

0.037

0.158

0.009

5.873

Serine

12.489

0.005

0.040

0.237

0.007

2.853

Cadaverine

13.754

0.005

0.037

0.058

0.002

4.139

Glucose

20.692

0.003

0.013

10.506

0.475

4.523

Sucrose

21.182

0.003

0.014

0.195

0.007

3.578

Stearic acid

22.794

0.004

0.016

0.063

0.002

2.584

Abbreviation:RSD: relative standard deviation.SD:standard deviation.

Tab S3 Screen in CSF marker between control and cerebral IR

Compound

Log2(FC)

Log10(p)

Compound

Log2(FC)

Log10(p)

Lysine

-5.432

2.703

Arabitol

-1.890

5.254

Propanoic acid

-5.066

9.415

2(3H)-Furanone

-1.737

1.302

13-Docosenamide

-3.794

5.836

Dodecanoic acid

-1.338

2.119

Sucrose

-3.222

5.474

n-Butylamine

-1.288

1.543

Gluconic acid

-3.100

2.913

Acetic acid

-0.821

4.083

2,3,4-TBA

-3.036

7.270

1,2,3-PA

-0.811

2.231

2-monopalmitin

-2.887

7.536

Hexadecanoic acid

-0.712

2.109

D-Fructose

-2.881

6.850

ethane-1,2-diol

0.831

2.794

Dulcitol

-2.782

8.923

Pantothenic acid

0.942

1.452

Threitol

-2.717

6.489

Ethanedioic acid

1.161

1.905

Lyxose

-2.614

11.446

Glucose

1.650

7.664

2-Monostearin

-2.573

1.932

2-Thenaldehyde

2.490

3.221

Ribonic acid

-2.561

11.687

Alanine

2.689

3.122

Glycerol

-2.540

9.927

borate

2.698

10.277

Galactopyranose

-2.540

1.811

Tyrosine

2.744

8.730

Ethane

-2.514

13.378

Ribose

2.914

8.069

Cadaverine

-2.490

1.939

Erythrotetrofuranose

3.032

7.199

Dimethylamine

-2.421

13.737

Butanoic acid

3.193

0.656

3-Buten-2-ol

-2.390

15.900

Creatinine

3.555

6.126

Sorbitol

-2.323

3.251

Serine

3.577

6.291

Oleic acid

-2.309

1.387

Pentane

3.772

6.677

D-Ribo-Hexitol

-2.050

1.723

1,4-Butanediol

4.049

6.168

Glycine

-2.007

3.464

Galactose

4.068

1.691

1,2-Butanediol

-1.957

1.993

Tab S4 The candidate markers of CSF between cerebral IR and Cop treated.

Compound

Log2(FC)

-Log10(p)

Ribose

Stearic acid

Glycerol

2,3,4-TBA

Glycine

Oleic acid

-2.502

-2.432

2.742

0.587

2.852

4.170

4.533

1.830

3.095

1.435

1.834

4.674

Tab S5 Identification in CSF markers between control, cerebral IR and Cop treated groups

Compounds

Control vs Cerebral IR

Cerebral IR vs Cop treated

Log2(FC)

-Log10(p)

Log2(FC)

-Log10(p)

Ribose

Glycerol

2,3,4-TBA

Glycine

Oleic acid

2.914

-2.540

-3.036

-2.007

-2.309

8.069

9.927

7.270

3.464

1.387

-2.502

2.742

0.587

2.852

4.170

4.533

3.095

1.435

1.834

4.674

  1. I don't have important observations, just two comments:
  • Definition of CSF is missing in the introduction (line 72), it only appears in the abstract
  • In the description of results obtained in Figure 4, the authors should indicate the panel to facilitate comprehension.

Response: Thanks for your suggestion. (1) The definition of CSF has been written in the introduction section and highlighted in the manuscript (“CSF is a biological fluid that circulates in the brain, and it is in direct contact with the extracellular space of the brain, and its changes reflect different functional states of the brain”). (2) The description of results obtained in Fig 4 add in the section of result as follow: “There were 47 differential metabolites were discovered between IR and control groups in CSF, including 16 increased (ribose, ethanedioic acid and butanoic acid etc) and 31 decreased(glycine, oleic acid, gluconic acid and cadaverine, etc), otherwise there were 6 differential metabolites in CSF (4 increased(oleic acid, glycerol, glycine and 2,3,4-trihydroxybutyric acid) and 2 decreased(ribose and stearic acid)) after Cop-treated.” and a marker list between control, cerebral IR and Cop treated was attached in Fig S3.

  1. Conclusion should be changed before methods

Response: Thanks for your advice. However, the format order of 《Molecules》 is introduction-results-discussion-materials and methods-conclusion.

  1. Although in general, the authors use Standard English, there are some words that are not used properly. I suggest reviewing carefully the text. For example:

Line 82: “the cerebral ischemia size was apparently increased in cerebral…” The common form to refer to the damaged area is “infarct area” instead of “ischemia size”. The authors found statistical differences between control and ischemic groups, so, it is not necessary to say “apparently”.

Line 85: “…to investigate whether the protective effect...”. The use of "whether" is incomplete in the sentence.

Lines 142 and 144, have similar errors.

Response: Thanks for your comments. These mistakes have been corrected, for example: we used “infract size” instead of “ischemia size”. 
